# PackNets: A Variational Autoencoder-Like Approach for Packing Circles in Any Shape[*]

## Abstract

The problem of packing smaller objects within a larger one has long been of interest. In this work, we employ an encoder-decoder architecture, parameterized by neural networks, for circle packing. Our solution consists of an encoder that takes the index of a circle as input and outputs a point, which is then transformed by a constraint block into a valid center within the outer shape. A perturbation block perturbs this center while ensuring it remains within the corresponding radius, and the decoder estimates the circle's index based on the perturbed center. The functionality of the perturbation block is akin to adding noise to the latent space variables in variational autoencoders (VAEs); however, it differs significantly in both the method and purpose of perturbation injection, as we inject perturbation to push the centers of the circles sufficiently apart. Additionally, unlike typical VAEs, our architecture incorporates a constraint block to ensure that the circles do not breach the boundary of the outer shape. We design the constraint block to pack both congruent and non-congruent circles within arbitrary shapes, implementing a scheduled injection of perturbation from a beta distribution in the perturbation block to gradually push the centers apart. We compare our approach to established methods, including disciplined convex-concave programming (DCCP) and other packing techniques, demonstrating competitive performance in terms of packing density—the fraction of the outer object's area covered by the circles. Our method outperforms the DCCP-based solution in the non-congruent case and approaches the best-known packing densities. To our knowledge, this is the first work to present solutions for packing circles within arbitrary shapes.

## 1 Introduction

Packing problems are commonly encountered in various domains of study. These problems involve packing smaller objects into a larger one to achieve objectives such as maximizing packing density. The circle packing problem, a specific instance of packing problems, focuses on packing circles within a larger shape. This non-convex problem arises in various applications Zhang et al. (2013), such as packing circular objects in a box Castillo et al. (2008), and in fields including nanotechnology, telecommunications Cover & Thomas (2006), the oil and automobile industries Wang et al. (2002), forestry Hifi & M'Hallah (2009), and social distancing Bortolete et al. (2022). These problems are straightforward to state but difficult to solve, even approximately. For instance, Kepler's conjecture, which concerns sphere packing in three-dimensional Euclidean space, was proved only recently Hales et al. (2015).

In this paper, we focus on developing a method to find sub-optimal solutions to packing problems. The work in Jose et al. (2024) focused on packing circles of equal radii, i.e., congruent circles, within a large circle using an autoencoder architecture. In this work, we extend the method to pack circles of varying radii, i.e., non-congruent circles, into arbitrary shapes. Let the outer shape be centered at the origin of a Cartesian coordinate system, with the distance to its boundary at angle $\theta$ from the horizontal axis denoted by $b(\theta)$, for $\theta \in [0, 2\pi]$. Within this outer shape, we aim to pack $N \geq 1$ circles, indexed by $i \in \mathcal{N} = \{1, 2, \ldots, N\}$, with radii $r_i$ by finding the centers $\mathbf{c}_i$ for all $i \in \mathcal{N}$. This packing must satisfy two constraints: (1) no circle should extend outside the outer shape, and

---

[*]The code associated with this work can be accessed here: `https://github.com/oddjoobs/ICLR_CirclePacking`

(2) the overlap between any two circles must be zero or below a certain threshold. That is, we are interested in solving the following optimization problem.

$$\underset{\mathbf{c}_i \in \mathbb{R}^2}{\textbf{minimize}} \qquad O = \sum_{1 \leq i < j \leq N} \text{overlap}(\epsilon_{ij}), \tag{1a}$$

$$\textbf{subject to} \qquad ||\mathbf{c}_i + r_i \begin{pmatrix} \cos(\psi) \\ \sin(\psi) \end{pmatrix}||_2 \leq b(\theta), \ \forall \psi, \theta \in [0, 2\pi], \ \forall i \in \mathcal{N}, \tag{1b}$$

$$r_i + r_j - ||\mathbf{c}_i - \mathbf{c}_j||_2 \leq \epsilon_{ij}, \ \forall i, j \in \mathcal{N}, i \neq j, \tag{1c}$$

where equation 1b ensures that no circle extends outside the outer shape, and equation 1c ensures that the overlap between any two circles is either zero or below a specified threshold. If the outer shape is a larger circle centered at the origin with radius $R$, i.e., $b(\theta) = R$ for all $\theta \in [0, 2\pi]$, equation 1b specializes to $||\mathbf{c}_i||_2 + r_i \leq R, \ \forall i \in \mathcal{N}$. Similarly, if the outer shape is a regular polygon, where angle to a vertex from the origin is zero, the distance $b(\theta)$ from the origin to the nearest edge of a regular polygon inscribed in a circle with circumradius $R$ and $n$ sides, as a function of the angle $\theta$, is given by $b(\theta) = R \cos(\pi/n) / \cos(\theta \mod (2\pi/n) - \pi/n)$, where $\theta$ is the angle from the center of the polygon, $\pi/n$ is half the central angle between two adjacent vertices, and $\theta \mod 2\pi n$ represents the angular position relative to the nearest vertex.

Our contribution is the development of a solution utilizing an encoder-decoder approach, akin to Jose et al. (2024), but with significant modifications. This approach aims to determine the coordinates of the centers of smaller circles within an outer shape while minimizing overlap. Unlike Jose et al. (2024), we implement several architectural modifications and extend our focus to packing non-congruent circles within various outer shapes. The outer shapes considered include circles, regular polygons (such as squares and regular pentagons), and arbitrary shapes.

The circle packing problem, defined by the no-overlap condition, has been extensively studied using various techniques, including non-linear programming, stochastic search, heuristics, and neural gas methods Jose et al. (2024). Non-linear programming approaches, such as the stochastic item descent method He et al. (2020) and the formulation by Graham et al. (1998), focus on maximizing pairwise distances between points within a unit circle. Stochastic search methods, proposed by Akiyama et al. (2003), involve initially scattering points and adjusting their positions until they settle without overlapping. This method has been further refined in Graham et al. (1998) and applied to packing circles in squares, as seen in Boll et al. (2004) and Szabó et al. (2007). Heuristic algorithms, which can be classified into construction and optimization methods, have also been explored Huang & Ye (2011); fu Zhang & sheng Deng (2005); Grosso et al. (2010); Zeng et al. (2016). Construction algorithms place circles incrementally, while optimization algorithms refine an initial solution. Additionally, the neural gas method Pospíchal (2015) uses clustering of random points to determine circle centers.

## 2 PROPOSED ENCODER-DECODER APPROACH

The architecture for packing equal-sized circles into a larger circle in Jose et al. (2024) consists of an encoder, normalization layer, perturbation layer, and decoder, where encoder and decoder layers are parameterized by neural networks. We adopt a similar architecture for packing circles within arbitrary shapes but introduce the following modifications: the encoder neural network is significantly reduced in size, and batch normalization layers are added, along with Tanh activations to ensure that the output points lie within a unit square. The normalization layer is replaced by a new *constraint block* consisting of learnable parameters, which provides greater freedom for the circle center to move within the outer shape while satisfying the constraint equation 1b. The perturbation layer now uses a beta distribution to introduce noise, where we initially perturb the center by small amounts and gradually increase the perturbation as the packing process progresses. Additionally, the number of perturbations is increased. Finally, the decoder is expanded, and batch normalization is added.

### 2.1 MODEL ARCHITECTURE

In this subsection, we describe each block in detail, and a block diagram is presented in Fig. 1.

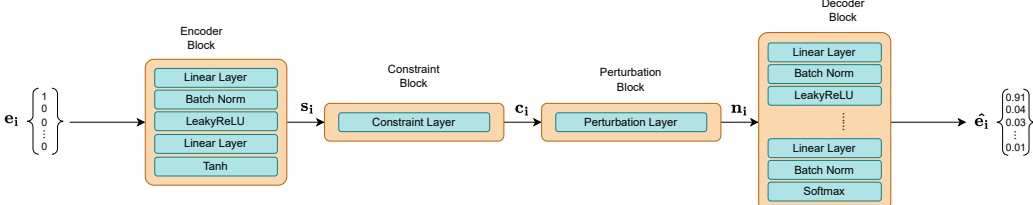

Figure 1: A block diagram showing various blocks in the proposed approach for packing circles within arbitrary shapes.

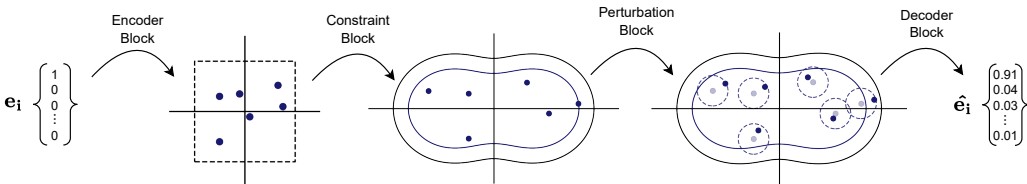

Figure 2: An illustration of typical outputs of different blocks in packing circles within arbitrary shapes.

**Encoder Block**  The encoder, $f_\Theta$, parameterized by a neural network $\Theta$, takes as input a one-hot indicator vector $\mathbf{e}_i \in \{0, 1\}^N$, where the $i$th element is one and the remaining elements are zeros, representing the $i$th circle to be packed, for $i \in \mathcal{N}$. It outputs a vector $\mathbf{s}_i$, i.e., $\mathbf{s}_i = f_\Theta(\mathbf{e}_i)$, which is then transformed by the constraint block into the center $\mathbf{c}_i$ for Circle $i \in \mathcal{N}$. The encoder is composed of two linear layers, a batch normalization layer, and layers with Leaky ReLU activations, while the final layer uses the Tanh activation function to output a vector $\mathbf{s}_i \in [-1, 1]^2$, ensuring that $\mathbf{s}_i$ lies within the unit square, centered at the origin, as shown in Fig. 2.

**Constraint Block**  This block transforms $\mathbf{s}_i$ to a feasible $\mathbf{c}_i$ as follows. First, we normalize $\mathbf{s}_i$ by dividing by its 2-norm, resulting in $\mathbf{s}'_i = \mathbf{s}_i / \|\mathbf{s}_i\|$, which means that $\mathbf{s}'_i$ lies within a unit circle centered at the origin. Now, we transform $\mathbf{s}'_i$ to lie anywhere within the outer shape. For this, we allow $\mathbf{s}'_i$ to be rotated as $\mathbf{s}''_i = \mathbf{R}(\lambda_i)\mathbf{s}'_i$, where the rotation matrix $\mathbf{R}(\lambda_i)$ is defined as $\mathbf{R}(\lambda_i) = \begin{pmatrix} \cos(\lambda_i) & -\sin(\lambda_i) \\ \sin(\lambda_i) & \cos(\lambda_i) \end{pmatrix}$, where $\lambda_i$ is the parameter to be learned during the packing process. Along the direction of $\mathbf{s}''_i$ (say $\angle \mathbf{s}''_i = \phi$), let $l_i(\phi)$ be the maximum $\|\mathbf{c}_i\|$ that ensures equation 1b is satisfied. Now the new rotated point $\mathbf{s}''_i$ is multiplied by $l_i(\phi)$ and $\delta_i$, where $\delta_i$ is a learnable parameter value lying between $[0, 1]$. By varying $\delta_i$, the center can be shifted along the line with length $[0, l_i(\phi)]$, along angle $\phi$. This final vector obtained is interpreted as the center of Circle $i$, $\mathbf{c}_i$. Thus, this layer allows for rotating and scaling each point of the encoder output such that the centers stay inside the outer shape. In this, the learnable parameters are $\lambda_i \in [0, 2\pi]$ and $\delta_i \in [0, 1]$, which are enforced by normalizing a vector and by using Sigmoid functions, respectively.

We now describe how we obtain $l_i(\phi)$, which indicates how far we can extend in the direction of $\phi$ without crossing the boundary of the outer shape. For simpler outer shapes, such as a circle, we have $l_i(\phi) = R - r_i$. In general, there may not be a closed-form expression for $l_i(\phi)$. In such cases, one can utilize a lookup table containing a finite number of points. However, if the angle $\phi$ does not appear in the lookup table, it becomes unclear how to proceed. In this work, we employ a radial basis function network to estimate $l_i(\phi)$ for any given $\phi$. This network is trained on pairs of $(\phi, l_i(\phi))$ that we obtain as follows: We select an arbitrary angle $\theta$ (see Fig. 3) and draw a line at this angle that intersects the boundary of the outer shape. At this intersection point, we draw a tangent and then a perpendicular to the tangent, tracing it for a length of $r_i$ within the shape. The length of the resultant point gives us $l_i(\phi)$, and the angle is $\phi$. We repeat the above process to obtain a sufficient number of $\theta$ values and train the resultant data to obtain a radial basis function network model that maps $\phi$ to $l_i(\phi)$. The current method expects the curve $\{l_i(\phi) \mid \forall \phi \in [0, 2\pi]\}$ should not self-intersect.

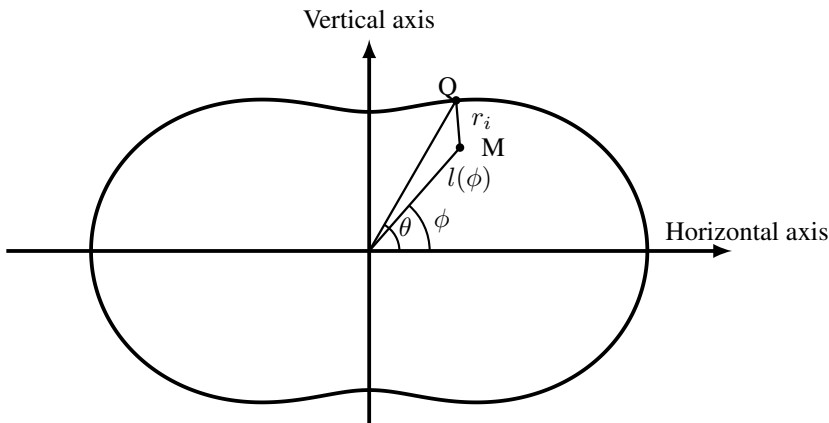

Figure 3: In this work, we need $l_i(\phi)$, which is the maximum distance from the center of Circle $i$ along the direction $\phi$ such that if the circle is centered at that point, it does not breach the boundary in any direction. To obtain pairs of $(\phi, l_i(\phi))$, we select an arbitrary angle $\theta$ and draw a line at this angle that intersects the boundary of the outer shape at point $Q$. At this intersection point, we draw a tangent and then a perpendicular to the tangent, tracing it for a length of $r_i$ within the shape (i.e., line segment $QM$). The length of the resultant point $M$ gives us $l_i(\phi)$, and the angle is $\phi$.

**Perturbation Block**  We perturb the center $\mathbf{c}_i$ by adding a random vector $\mathbf{w}$, ensuring that $\|\mathbf{c}_i - \mathbf{w}\|_2 \leq r_i$ for all $i \in \mathcal{N}$. To generate $\mathbf{w}$, we first sample two independent scalars from the uniform distribution over $[0, 1]$, stack them as a vector, and normalize it to obtain the unit vector $\mathbf{u}_i$. This provides a random direction. Next, we sample a scalar $\rho$ from the Beta distribution Beta$(\alpha, \beta)$, and scale it by $r_i$. Since the Beta distribution's support is $[0, 1]$, $\rho r_i$ ensures the perturbation magnitude lies within $[0, r_i]$. Finally, we set $\mathbf{w} = \mathbf{u}_i \rho r_i$, ensuring that the perturbation remains within the desired radius. The output of the perturbation block is $\mathbf{n}_i = \mathbf{c}_i + \mathbf{w}$.

We do not use fixed values for $\alpha$ and $\beta$; instead, we change them according to a schedule as the packing process progresses. This approach ensures that initially, we sample points closer to the center, allowing the circles to gradually push away. In later stages, when the circles are no longer in close proximity, we aim to sample points farther from the center. The rationale is that sampling points from regions where circles intersect increases the difficulty for the decoder, making accurate predictions harder. To counter this, the encoder pushes the circles apart, thereby reducing the loss and improving packing efficiency. Additionally, sampling more points per circle can accelerate the model's learning process. This scheduling is explained in a later section when detailing the training process.

The functionality of this block is akin to adding noise to the latent space variables in variational autoencoders; hence, we refer to this work as a variational autoencoder-like approach. However, it differs in both the method and purpose of perturbation injection, as we inject perturbation to sufficiently separate the centers of the circles. Additionally, unlike typical VAEs, our architecture incorporates a constraint block to ensure that the circles do not breach the boundary of the outer shape.

**Decoder Block**  This block is composed of many linear layers, batch normalization layers, and ReLU activation functions (except for the last layer, which uses Softmax). The perturbed point $\mathbf{n}_i = \mathbf{c}_i + \mathbf{w}$ is passed to the decoder, $g_\Phi$, which makes a prediction about the index the point belongs to. Specifically, it outputs a probability vector $\hat{\mathbf{e}}_i$, indicating the likelihood that the perturbed point was generated from $\mathbf{e}_i$.

## 2.2 Loss Function and Training Process

The parameters of the encoder, $\Theta$, parameters of the constraint block, $\lambda_i$ and $\delta_i$, and the decoder, $\Phi$, are optimized to minimize $(1/N) \sum_{i=1}^{N} \mathbb{E}_{\mathbf{W}} \left[ \mathbb{I}_{\hat{i} \neq i} \right]$, where $\mathbb{I}$ is an indicator variable that equals

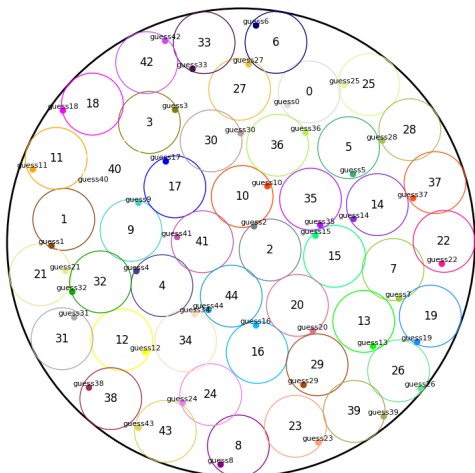

Figure 4: An illustration of packing for $N = 45$ circles, each of radius $r_i = 0.13204959425$ for all $i \in \mathcal{N}$, within an outer circle of unit radius. The numbers within a circle represent the index of the circle. The output of the encoder and constraint layer corresponds to the center of the $i^{\text{th}}$ circle plotted (center not shown). The point of the same color as the circle boundaries near the $i^{\text{th}}$ circle is a perturbed center of the circle. The decoder takes these perturbed points as inputs and outputs an index, $\hat{i}$, which is referred to as "guess $\hat{i}$".

1 if $\hat{i} \neq i$ and 0 otherwise, and $\hat{i}$ is the index of the location of the maximum value of $\hat{\mathbf{e}}_i$, i.e., $\hat{i} = \arg\max \hat{\mathbf{e}}_i$. This approach yields a circle packing solution that avoids overlaps whenever possible and minimizes their extent in cases where overlaps are unavoidable. In this work, we use the average cross-entropy loss between $\mathbf{e}_i$ and $\hat{\mathbf{e}}_i$ for $i \in \mathcal{N}$ as a proxy for $(1/N) \sum_{i=1}^{N} \mathbb{E}_{\mathbf{W}} \left[ \mathbb{I}_{\hat{i} \neq i} \right]$ and minimize it using standard techniques employed to train an autoencoder.

For training, as mentioned, we create a scheduler to control the noise injection in a systematic manner. Our scheduler works as follows: we begin with arbitrary values for $\alpha$ and $\beta$, such that $\alpha < \beta$. For our case, we initially set $\alpha = 2$ and $\beta = 8$, which results in a probability mass closer to zero, implying that the center is perturbed only slightly, keeping it near the original position. We train the network with these values for a specified number of epochs while monitoring the empirical packing density. If the packing density stagnates, meaning it does not increase over several epochs (based on a patience hyperparameter), we adjust the parameters as follows: $\alpha \leftarrow \alpha + 0.28$ and $\beta \leftarrow \beta - 0.46$. This adjustment shifts the probability mass away from zero, allowing for larger perturbations. We continue this process until $\beta$ reaches 2, at which point we stop modifying it, while $\alpha$ can keep increasing (in our case, we cap it at 16). The rate at which the beta distribution is modified, as well as the conditions for stagnation, are additional hyperparameters that can be fine-tuned to achieve better packing results.

## 3 RESULTS

To evaluate the performance of our approach, we consider the following packings: congruent circles in a circle, square, and pentagon, as well as non-congruent circles in a circle, square, and an arbitrary shape. The outer circle is assumed to have a radius of 1 unit, the outer square a side length of 1 unit, and the outer regular pentagon a circumradius of 1 unit. For concreteness, we define the arbitrary shape by $b(\theta) = 1 + \cos^2(\theta)$ for $\theta \in [0, 2\pi]$.

This work focuses on determining the centers of the circles, given the radii of the smaller circles to be packed. Except for the case where the outer shape is arbitrary, the radii of the circles that can be packed without overlap are obtained from Packomania (2024). Specifically, the circles are indexed from smallest to largest, i.e., the circle with index 1 has the smallest radius, and the circle with index $N$ has the largest radius. In the case of congruent circles, the radius of each circle is $r_i = r_N$,

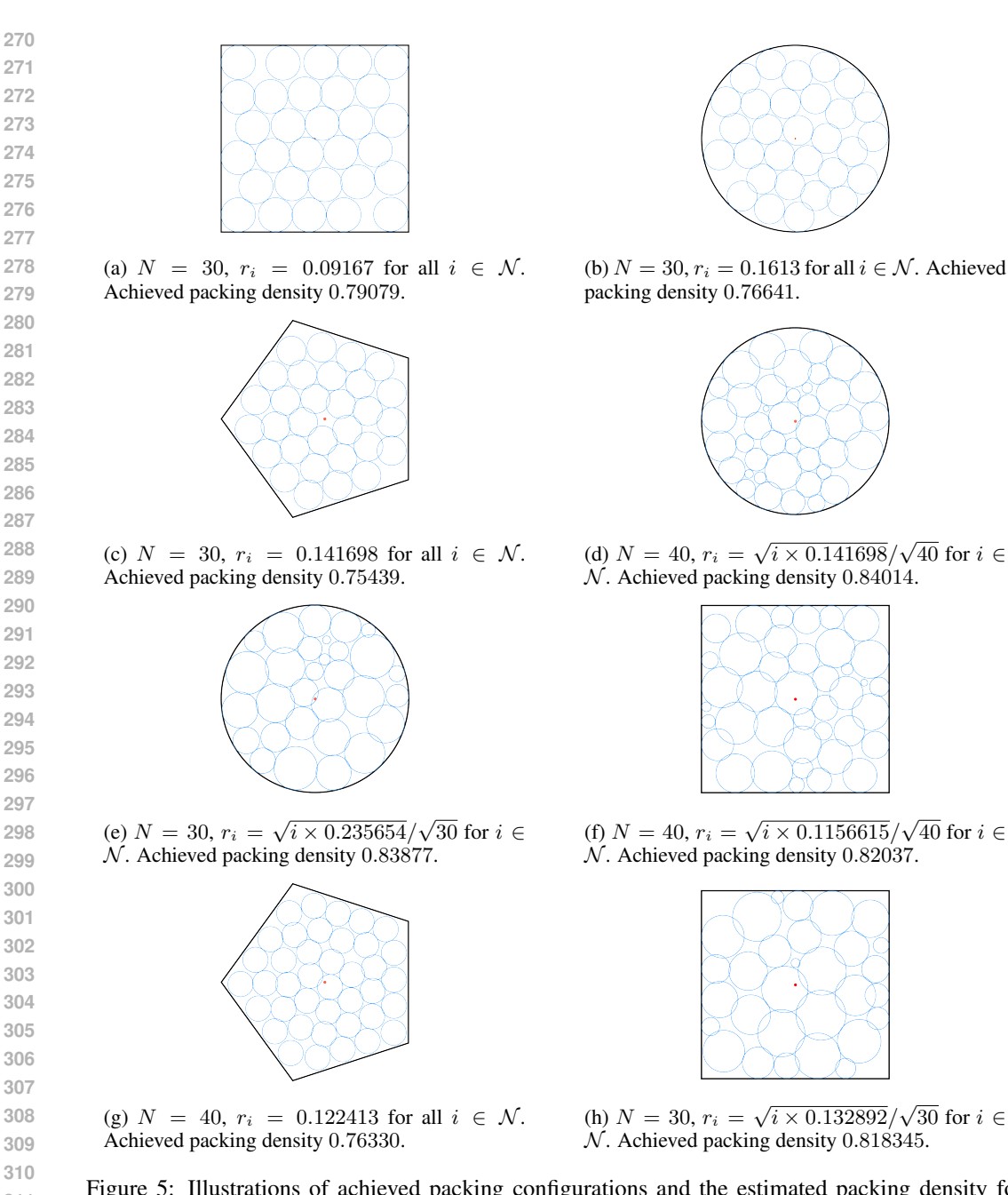

(a) $N = 30$, $r_i = 0.09167$ for all $i \in \mathcal{N}$. Achieved packing density 0.79079.

(b) $N = 30$, $r_i = 0.1613$ for all $i \in \mathcal{N}$. Achieved packing density 0.76641.

(c) $N = 30$, $r_i = 0.141698$ for all $i \in \mathcal{N}$. Achieved packing density 0.75439.

(d) $N = 40$, $r_i = \sqrt{i \times 0.141698}/\sqrt{40}$ for $i \in \mathcal{N}$. Achieved packing density 0.84014.

(e) $N = 30$, $r_i = \sqrt{i \times 0.235654}/\sqrt{30}$ for $i \in \mathcal{N}$. Achieved packing density 0.83877.

(f) $N = 40$, $r_i = \sqrt{i \times 0.1156615}/\sqrt{40}$ for $i \in \mathcal{N}$. Achieved packing density 0.82037.

(g) $N = 40$, $r_i = 0.122413$ for all $i \in \mathcal{N}$. Achieved packing density 0.76330.

(h) $N = 30$, $r_i = \sqrt{i \times 0.132892}/\sqrt{30}$ for $i \in \mathcal{N}$. Achieved packing density 0.818345.

Figure 5: Illustrations of achieved packing configurations and the estimated packing density for packing circles within circles and regular polygons.

while for non-congruent circles, $r_i = \sqrt{i \times r_N}/\sqrt{N}$, for all $i \in \mathcal{N}$, where $r_N$ is the radius of the circle with index $N$ and represents the largest radius. The $r_N$ values considered are mentioned in the specific results. For arbitrary outer shapes, the radii of the inner circles are chosen randomly, as detailed in the specific results.

We compare our results with the disciplined convex-concave programming (DCCP)-based results from Shen et al. (2016) and the best-known results from Packomania (2024). Our performance metric is packing density, defined as the fraction of the total area occupied by the packed circles. For both the DCCP-based solution and our solution, we estimate the packing density using Monte Carlo sampling, where a large number of points are randomly generated within the larger object. The number of overlapping points is then computed to determine the packing density, similar to the

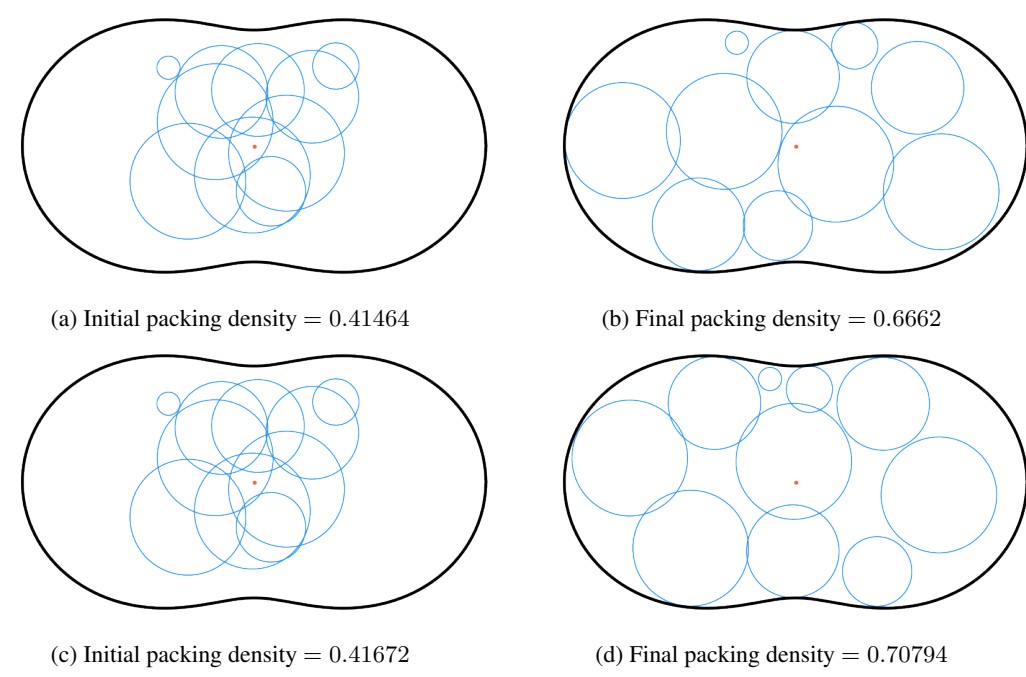

(a) Initial packing density $= 0.41464$        (b) Final packing density $= 0.6662$

(c) Initial packing density $= 0.41672$        (d) Final packing density $= 0.70794$

Figure 6: An illustration of circles in a shape with boundary $b(\theta) = 1 + \cos^2(\theta)$ for $\theta \in [0, 2\pi]$ with the number of circles $N = 10$. Four inner circles have a radius of $0.5$, three have a radius of $0.4$, and one each has a radius of $0.3$, $0.2$, and $0.1$. Fig. 6b shows the final packing corresponding to the initial packing in Fig. 6a, and similarly, Fig. 6d corresponds to Fig. 6c.

approach in Jose et al. (2024). For the best-known case from Packomania (2024), we report the packing density as provided, which we believe is theoretically computed.

In Fig. 4, we illustrate the packing for $N = 45$ circles, each with radius $r_i = 0.13204959425$ for all $i \in \mathcal{N}$, within an outer circle of unit radius. From the figure, we observe that even when the perturbed center, $\mathbf{c}_i + \mathbf{w}$, is very close to the boundary of the inner circle, the decoder model is able to correctly classify it as belonging to the correct circle with center $\mathbf{c}_i$, which was output by the encoder.

In Fig. 5, we show the achieved packing configurations and the estimated packing density obtained by the proposed encoder-decoder-based approach for packing circles within a circle, square, and regular pentagon. The packing obtained is visually close to the best-known packing configurations reported in Packomania (2024) under these settings. Additionally, the packing density achieved by our approach is also close to the best-known packing densities reported in Packomania (2024).

In Fig. 6, we compare the initial and final packing of non-congruent circles within a shape defined by the boundary $b(\theta) = 1 + \cos^2(\theta)$ for $\theta \in [0, 2\pi]$. The initial packing is achieved using the default initialization of neural network weights from PyTorch (2024). In the constraint block, the parameter $\lambda_i$ is initialized as described earlier, while $\delta_i$ follows a Gaussian random variable with mean $-0.75$ and variance $1$. The figure illustrates that the proposed method ensures all circles remain within the larger circle's boundary and overlap minimally. Additionally, we observe an increase in packing density compared to the initial configuration. In this case, there are no benchmark methods available to compare the final packing density against.

In Table 1, we present the packing densities obtained from a comparison of our approach, the DCCP-based solution from Shen et al. (2016), and the best-known packing configurations from Packomania (2024). Our approach demonstrates comparable performance to DCCP, with both methods achieving packing densities that are very close to the best-known configurations, regardless of whether we are packing congruent or non-congruent circles. The packing density obtained by the proposed method is greater than that of the DCCP-based method for the non-congruent case.

Table 1: Comparison of packing densities for packing inner circles. In the case where the inner circles are congruent, the radius of the inner circle with index $i$ is $r_i = r_N$. For the non-congruent case, $r_i = \sqrt{i \times r_N}/\sqrt{N}$, where $r_N$ is the radius of the circle with index $N$ and represents the largest radius of the inner circle.

| Outer Shape, Congruency of Inner Circles | $N$ | $r_N$ | Packing Density | | |
|---|---|---|---|---|---|
| | | | Our Approach | DCCP in Shen et al. (2016) | Best Known Packomania (2024) |
| Circle, Congruent | 10 | 0.26225 | 0.6875 | 0.685184 | 0.68779 |
| Circle, Congruent | 20 | 0.195224 | 0.76205 | 0.7621 | 0.76224 |
| Circle, Congruent | 30 | 0.161349 | 0.76714 | 0.7714 5 | 0.78100 |
| Circle, Congruent | 40 | 0.140373 | 0.7781 | 0.7831 | 0.78818995 |
| Square, Congruent | 10 | 0.1482043 | 0.6897 | 0.685184 | 0.690035 |
| Square, Congruent | 20 | 0.111382 | 0.773 | 0.7419 | 0.779493 |
| Square, Congruent | 30 | 0.091671 | 0.7911 | 0.7800 | 0.792019 |
| Square, Congruent | 40 | 0.079186 | 0.7411 | 0.7879 | 0.787979 |
| Pentagon, Congruent | 10 | 0.230721 | 0.671889 | 0.697512 | 0.70336 |
| Pentagon, Congruent | 20 | 0.169279 | 0.74059 | 0.752128 | 0.75725 |
| Pentagon, Congruent | 30 | 0.141698 | 0.7710 | 0.78023 | 0.79589 |
| Pentagon, Congruent | 40 | 0.122413 | 0.7533 | 0.791434 | 0.79199 |
| Circle, Non-Congruent | 10 | 0.3808381 | 0.78691 | 0.76335 | 0.79770 |
| Circle, Non-Congruent | 20 | 0.2832029 | 0.82003 | 0.81261 | 0.842140 |
| Circle, Non-Congruent | 30 | 0.235654 | 0.83877 | 0.83257 | 0.86076 |
| Circle, Non-Congruent | 40 | 0.205996 | 0.84014 | 0.84137 | 0.86990 |
| Square, Non-Congruent | 10 | 0.216258 | 0.7688 | 0.766325 | 0.808086 |
| Square, Non-Congruent | 20 | 0.160535 | 0.8013 4 | 0.781968 | 0.850123 |
| Square, Non-Congruent | 30 | 0.1328929 | 0.818345 | 0.818335 | 0.8599747 |
| Square, Non-Congruent | 40 | 0.1156615 | 0.82037 | 0.83386 | 0.8615524 |

## 4 CONCLUSIONS AND FUTURE WORK

In this work, we employed an encoder-decoder architecture for packing circles within arbitrary shapes. The architecture consisted of an encoder that took the index of a circle as input and output a point, which was then transformed by a constraint block into a valid center within the outer shape. A perturbation block perturbed this center while ensuring it remained within the corresponding radius, and the decoder estimated the circle's index based on the perturbed center. The constraint block was designed to accommodate both congruent and non-congruent circles, implementing a scheduled injection of perturbation from a beta distribution to gradually push the centers apart.

Our approach demonstrated competitive performance compared to established methods, including DCCP and best-known configurations, in terms of packing density, which we computed by scattering points within the outer shape and counting the fraction that lay within the area covered by the smaller circles. We illustrated that the method successfully packed circles within a unit circle, accurately classifying perturbed centers close to the boundary. Visual results showed that the packing configurations were similar to the best-known configurations for various shapes, achieving packing densities that approached the best-known. Notably, the proposed method outperformed DCCP in packing non-congruent circles, achieving higher packing densities. To our knowledge, this was the first work to present solutions for packing circles within arbitrary shapes.

Future work will focus on extending the method to pack any inner shape, which makes enforcing constraints difficult. This will require modifications to the constraint and perturbation blocks. Additionally, the perturbation scheduler can be further optimized, and enhancements can be made to

the initialization process. Increasing the complexity of the constraint and encoder layers will also be explored to achieve better packing results.

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
