# OpenReview forum: "PackNets: A Variational Autoencoder-Like Approach for Packing Circles in Any Shape"
_ICLR.cc/2025/Conference — Submitted to ICLR 2025_

### Official Review · Reviewer_5Ar9 · 2024-10-24

**Soundness:** 2
**Presentation:** 3
**Contribution:** 1
**Rating:** 3
**Confidence:** 4

**Summary:**

This paper introduces a method of solving 2D circle packing problems by jointly training an encoder that predicts the center of a given circle to pack and a decoder that predicts the circle a given point belongs to (during training this point is generated by a random perturbation from the center determined by the radius). The encoder and decoder are parameterized by neural networks. This is motivated by the intuition that if the encoder-decoder is trained to a low loss, the encoder will predict centers that the decoder can distinguish even up to a perturbation by the radius of the circle, thus generating a valid packing. This method is then tested on instances of packing circles of fixed radius into various 2D shapes.

**Strengths:**

This paper explores a new approach to solve packing problems, by restating the problem as training an encoder-decoder, potentially using ideas and architectures from the vast VAE literature. The writing is clear and it is easy to understand the architectural choices in the design of the neural network.

**Weaknesses:**

Although the approach is new and seems promising, in my opinion this paper does not meet the standards of acceptance, as 1) it only solves a problem of limited scope (2D circle packing), 2) it violates the non-overlap constraint that is essential to this problem and is respected by all previous approaches, 3) the evaluation metrics are not comparable to other approaches due to the presence of overlaps.

1. By far the biggest weakness in this paper is that the circles in a packing generated by the algorithm can overlap with each other. This is different from the other approaches which considers packings with strictly non-overlapping circles. This relaxation of the non-overlapping constraint seems to be inherent to this encoder-decoder approach where the penalty for overlap is proportional to the overlap area, thus making it hard to eliminate all overlaps.

2. This also calls into question whether the comparisons of packing ratios with previous approaches are fair, as previous approaches do not allow overlaps, whereas the packings generated here do have overlaps.

3. It was claimed that this method handles arbitrary shapes, but in the paper these shapes are parameterized by a radial function $b(\theta)$. However, this parameterization limits the shapes that can be expressed. For example, shapes with holes in them such as an annulus cannot be captured with this parameterization. I would suggest that the authors reduce the scope of this claim.

4. The evaluation only reported individual run results and did not report any statistics. It would be stronger if this section reported statistics such as the average gap to the Packomania results, or the fraction of instances it achieved a better density over DCCP.

Minor comments:

1. Table 1 is hard to read, it would help to instead report the difference in packing density compared to the best known.

**Questions:**

1. The beta distribution is chosen to sample a perturbed point from the center. How does the perturbation distribution affect the training dynamics? Is training faster if points closer to the edges of the circle are more likely to be sampled during perturbation? Will there be fewer overlaps if points on the boundary of the circle are sampled with higher probability?

2. Since this approach generates overlapping circles, have you considered  generating packings with no overlaps by keeping the centers fixed and reducing the radius of circles until there are no overlaps?

3. It might be worth exploring applications of this method to higher-dimensional packing and covering problems, where approximate solutions are necessary and neural networks are better suited for parameterizing these spaces than more traditional methods.

---

### Official Review · Reviewer_Qugj · 2024-10-26

**Soundness:** 1
**Presentation:** 1
**Contribution:** 1
**Rating:** 1
**Confidence:** 5

**Summary:**

This work addresses the circle packing problem. It builds upon a previous work that searches for an optimal packing of the circles by learning an identity map between circle indices. This idea is loosely inspired by VAEs: the "encoder" places each circle in the shape. The "noising" then samples points from each circle. Finally, the "decoder" tries to identify to which circle each point belongs. This is only possible if the circles are non-overlapping -- hence the cross-entropy loss between input and output indices is minimized.
Several experiments are demonstrated on primarily circular and polygonal domains. Visual quality and packing density are used as evaluation metrics, suggesting comparable performance to one other method.

**Strengths:**

- The investigated application is well-motivated.
- The text is largely free of syntactic errors.
- I appreciate some figures, especially figure 2.
- The employed analogy to VAEs is an interesting viewpoint.

**Weaknesses:**

Unfortunately, the paper is extremely flawed in conception and execution.
- This work falls within the sub-domain of machine learning for optimization, see e.g. [1]. Fundamentally, these are optimization problems, where ML is employed to parameterize a solution or generalize across problem instances. Here, neither is done and I struggle to classify this as an ML paper -- it is primarily a problem-specific optimization algorithm. While two NNs ("encoder" and "decoder") are employed, it is not clear why these are even necessary:
  - instead of "learning the encoder", why not optimize for each center $s_i$ directly?
  - instead of "learning the decoder" to decode noisy circle samples back to indices, and computing cross-entropy to the input, why not pose a loss directly on the geometric circles without introducing additional variance and parameters?
While I do not see a principled reason for this, I am open to the possibility that I am wrong and this helps empirically -- but this must be demonstrated, e.g. using ablation studies, of which there are none overall.
- The evaluation is massively flawed. Even though the original statement is a constrained optimization problem, the employed metric is only the objective, i.e., the packing density, with the feasibility being completely ignored. Even visually it is obvious that the constraints are not satisfied as the circles often overlap significantly. As such, the metric and the results are misleading.
- Even with this flawed metric, the empirical results are at best comparable to one other method. However, there is no report of the runtimes. In such optimization problems, there is almost always a trade-off between optimality and runtime, which must be respected for a fair comparison.
- The method is stated to apply to arbitrary shapes, while the parametrization in line 50 and thus the constraint block applies to star-shaped domains only.
- There are several poor presentation choices, e.g., >10 significant digits reported, a poorly formatted table, or repeated references to solutions visually agreeing, while the Packomania solutions are never shown.
- The limitations are not discussed.

[1] Yoshua Bengio, Andrea Lodi, and Antoine Prouvost. Machine learning for combinatorial optimization: a methodological tour d’horizon. European Journal of Operational Research, 2021.

**Questions:**

I would invite the authors to clarify the necessity of using the NNs if they disagree with the above assessment.

Overall, I would strongly recommend addressing the aforementioned aspects (ablation studies, metric, feasibility, runtimes) and reconsidering whether an ML venue is the right fit for this work.

---

### Official Review · Reviewer_LKMJ · 2024-10-27

**Soundness:** 3
**Presentation:** 3
**Contribution:** 3
**Rating:** 6
**Confidence:** 2

**Summary:**

The paper proposes PackNets, a neural network-based method for packing circles within various shapes while maximizing packing density and minimizing overlaps. Inspired by variational autoencoders (VAEs), PackNets employs an encoder-decoder architecture with several unique components to address the complex circle packing problem in both congruent (same size) and non-congruent (different sizes) cases. PackNets uses an encoder to generate the initial positions of circles, which are then processed by a constraint block that ensures each circle center remains within the specified boundary. The perturbation block introduces controlled, scheduled "noise" to help separate the circles, a technique adapted from VAEs to improve spacing while maintaining circle boundaries. Finally, a decoder predicts the circle indices based on the positions generated, helping verify the integrity of the arrangement.  The authors tested PackNets across various shapes—circles, squares, regular polygons, and an arbitrary shape defined by a custom boundary function. The approach performed competitively, often achieving densities near the best-known packing results. For non-congruent circle packing, PackNets outperformed traditional methods like disciplined convex-concave programming (DCCP), achieving higher densities and more efficient layouts.

**Strengths:**

*  The method leverages an encoder-decoder structure which is new for packing problems, The method provides flexibility for adapting to various outer shapes which makes it useful for real-world applications.

* PackNets supports packing both congruent and non-congruent circles within arbitrary shapes, marking it as a significant advance over traditional packing methods that are often restricted to congruent circles and simple shapes.  The model achieved higher packing densities than DCCP (especially when packing circles of varying sizes) showing its robustness for complex arrangements.

* The perturbation block uses a gradual, scheduled approach to ensure circles are spaced optimally. Therefore, it helps improve packing density and reduce overlap without having computational overhead. The scheduled injection of perturbation gradually increases the distance between circles, balancing the need to maximize packing density while keeping overlaps minimal.

**Weaknesses:**

* While being effective for basic shapes, PackNets may require further adaptation for highly irregular or intricate boundaries.

* The success of PackNets depends on carefully tuning the parameters that control perturbation scheduling.

* The process may be computationally intensive for larger configurations, especially as the number of circles increases.

**Questions:**

1. How sensitive is PackNets’ performance to the specific values chosen for the perturbation schedule? How can we tune those parameters?

2. Could other loss functions be used to improve packing densities?

3. What specific real-world problems could benefit from PackNets?

---

### Official Review · Reviewer_XgxX · 2024-11-05

**Soundness:** 3
**Presentation:** 2
**Contribution:** 2
**Rating:** 5
**Confidence:** 3

**Summary:**

This paper proposes a variational autoencoder-like approach to find the sub-optimal solution to packing circles in arbitrary shapes while minimizing overlap. The paper introduces an encoder-decoder architecture that is parameterized by neural networks and consists of four blocks. The encoder block generates points for the circle positions. The constraint block enforces that the boundary of the outer shape is not breached at any point. The perturbation block applies controlled noise to push the circles further apart. Finally, the decoder provides a likelihood estimation of the circle’s index based on the perturbed point. Using packing density as a metric, the proposed approach is evaluated against the established disciplined convex-concave programming (DCCP) method as well as the best reported solutions on the Packomania platform. The packings that are considered are congruent circles in a circle, square, and pentagon, and non-congruent circles in a circle, square, and an arbitrary shape. The reported results show that the proposed approach outperforms both comparison models in the non-congruent case and has competitive results in the congruent case.

**Strengths:**

- Originality: The paper builds on previous work by Jose and colleagues (2024), who introduced a similar encoder-autoencoder architecture to packing equal-sized circles into a larger circle. However, by making various modifications to the original model, this approach can effectively be extended to build a model for packing circles within arbitrary shapes.

- Quality & Clarity: The proposed encoder-decoder approach, its architecture and components, as well as their functions are clearly explained and present a well thought out model that can successfully approximate solutions to complex packing problems.

- Significance: The new encoder-decoder approach shows great results for finding solutions to congruent and incongruent circle problems by reaching competitive packing density results as compared to the established DCCP model and approaching the best reported solutions from the Packomania platform. Importantly, this method can be used for packing of arbitrary shapes, which is a novel and significant contribution to the field.

**Weaknesses:**

1. The paper specifically states that the focus of the paper is to develop a method that finds sub-optimal solutions to packing problems. However, it is unclear why exactly the paper focuses on sub-optimal rather than optimal solutions. While there are certainly many domains that have applications for sub-optimal solutions to packing problems and do not need the stricter conditions of optimal non-overlapping solutions (e.g. due to heightened speed and computational efficiency; need for flexibility, etc.), the paper’s choice of focusing on sub-optimal solutions, its utility, and its implications should be clearly motivated.

2. A second lack of clarity arises from the paper not specifying how the developed solutions are sub-optimal. An optimal solution is usually defined by achieving maximal packing density, meeting all constraints (i.e. object dimensions, container boundaries, non-overlapping conditions), and achieving theoretical efficiency if the optimal configuration has been theoretically determined. It should be explicitly stated which one of these conditions is optimized for and which one not. One sub-optimality is clearly introduced by relaxing the second constraint, which allows for the overlap between any two circles to either be zero (classic optimal solution problem) or to be set below a certain threshold allowing for some overlap between the circles. Additional sub-optimalities should be clearly stated and discussed though.

3. The paper reports that the encoder-decoder approach outperforms the DCCP in the non-congruent cases. While this is numerically a true statement in all but one cases, the difference is often marginal (e.g.: 0.818345 vs. 0.818335). Model comparison of packing density should be valid based on numeric values as long as the model does not have any stochastic elements that could cause repeated runs to return different solutions in packing density. However, to my understanding, the encoder-decoder model does have multiple stochastic elements: 1) The controlled noise used to push the center position of circles apart in the perturbation block is sampled from a Beta distribution whose parameters change over the course of training. 2) The neural network parameters in the encoder, constraint, and decoder blocks are all initialized randomly. 3) The perturbation magnitude is based on a scheduler that adjusts the Beta distribution parameters during the training process. If different runs can lead to variable density packing values, how were the reported values in Table 1 chosen? What is the variance in density packing values over multiple runs? To facilitate a correct and more rigorous comparison, average packing density across multiple runs should be reported, including standard deviations to measure performance variability.

Also, is the stochasticity desired for generating diverse set of solutions / sampling the posterior and etc.?

4. Even though the strict non-overlap condition can be part of the second constraint, it does not seem to be enforced in any of the tested packing configurations. To further evaluate the model, it would be helpful to have a comparison of results for when the second constraint is set to zero.

5. The paper stresses multiple times that this is the first work to present solutions for packing circles within arbitrary shapes. Although this is named as one of the biggest contributions of the paper, there is no mention of why this is an important contribution and what its implications are.

**Questions:**

Addressing point 3 in the weaknesses is crucial for assessing the encoder-decoder approach’s performance and specifically for evaluating the claim that it outperforms the DCCP model in the non-congruent circle cases. Addressing all other points detailed in the weaknesses would significantly improve the clarity and significance of the paper.

---

### Meta-Review · Area_Chair_rQ3V · 2024-12-08

**Metareview:**

The paper adresses the problem of packing circles into arbitrary shapes, which is a meaningful problem in combinatorial optimization. They derive a VAE type architecture to solve for it. All the reviewers agreed about for the originiality and merits of this approach, but unanimously raised questions about the evaluation, and fairness of comparisons with exact approaches. In this light, I am recommending a reject decision, and I encourage the authors to further strengthen their work on the questions raised by reviewers.

**Additional Comments On Reviewer Discussion:**

No rebuttal was provided by authors. Most of the reviewers agreed for the final decision.

---

### Decision · Program_Chairs · 2025-01-22

Reject